# EfficientQAT: Efficient Quantization-Aware Training for Large Language Models

## Abstract

Large language models (LLMs) are crucial in modern natural language processing and artificial intelligence. However, they face challenges in managing their significant memory requirements. Although quantization-aware training (QAT) offers a solution by reducing memory consumption through low-bit representations with minimal accuracy loss, it is impractical due to substantial training resources. To address this, we propose Efficient Quantization-Aware Training (EfficientQAT), a more feasible QAT algorithm. EfficientQAT involves two consecutive phases: Block-wise training of all parameters (Block-AP) and end-to-end training of quantization parameters (E2E-QP). To the best of our knowledge, Block-AP is the first method to enable direct training of all parameters in a block-wise manner, reducing accuracy loss in low-bit scenarios by enhancing the solution space during optimization. E2E-QP then trains only the quantization parameters (step sizes) end-to-end, further improving the performance of quantized models by considering interactions among all sub-modules. Extensive experiments demonstrate that EfficientQAT outperforms previous quantization methods across a range of models, including base LLMs, instruction-tuned LLMs, and multimodal LLMs, with scales from 7B to 70B parameters at various quantization bits. For instance, EfficientQAT obtains a 2-bit Llama-2-70B model on a single A100-80GB GPU in 41 hours, with less than 3 points accuracy degradation compared to the full precision (69.48 vs. 72.41).

## 1 Introduction

Recent advancements in large language models (LLMs) (Touvron et al., 2023; Bubeck et al., 2023; Chiang et al., 2023; Xu et al., 2023a; Ying et al., 2024) have demonstrated impressive capabilities in diverse language tasks such as reasoning (Clark et al., 2018; 2019; Zellers et al., 2019), cognitive processing (Fu et al., 2023; Xu et al., 2023a), and agent-based applications (Qin et al., 2023a;b). However, these models are characterized by their extensive parameters, which pose significant challenges for memory footprint and bandwidth (Kim et al., 2023b; Xu et al., 2024a).

Quantization-aware training (QAT) is a highly effective quantization technique that minimizes quantization errors by incorporating quantization constraints during training. For example, BitNet b1.58 (Ma et al., 2024) can achieve nearly lossless ternary quantization. The precision of QAT is due to two main factors: 1) Fully trainable parameters allow for enough optimized space for gradient descent optimization; 2) End-to-end training accounts for interactions among all sub-modules in the models. Despite its performance benefits, QAT demands significant training resources, such as time and GPUs, as well as extensive training data. For instance, BitNet b1.58 requires retraining LLMs from scratch using the entire pre-trained dataset. Therefore, this approach is impractical for extremely large models and has only been verified on 3B models with 100B training tokens.

In optimizing quantization for LLMs, current methods emphasize either fine-grained reconstruction or reducing trainable parameters. While these approaches improve efficiency, they significantly degrade accuracy in low-bit scenarios. Mainstream post-training quantization (PTQ) methods (Lin et al., 2023; Frantar et al., 2022; Shao et al., 2023; Cheng et al., 2023b) focus on block-wise reconstruction (Li et al., 2021). They also restrict the optimization space to alleviate overfitting risk by only training rounding parameters (Nagel et al., 2020; Cheng et al., 2023b), clipping thresholds (Shao et al., 2023), or step sizes (Esser et al., 2019; Ding et al., 2023). However, these methods not only limit optimizable parameters but also overlook cross-block interactions, leading to notable accuracy degeneration

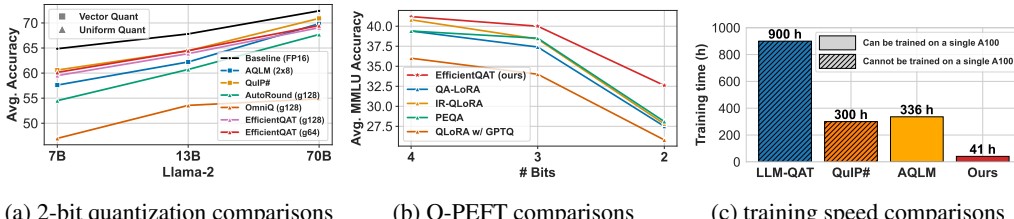

(a) 2-bit quantization comparisons     (b) Q-PEFT comparisons     (c) training speed comparisons

Figure 1: **(a)** EfficientQAT significantly surpasses existing uniform quantization methods, and is either superior to or comparable with vector quantization techniques. **(b)** EfficientQAT markedly outperforms existing Q-PEFT methods. **(c)** EfficientQAT can complete the QAT of 70B models on a single NVIDIA A100-80GB GPU with less time.

in low-bit scenarios, as shown in Figure 1a. Conversely, quantized parameter-efficient fine-tuning (Q-PEFT) methods (Dettmers et al., 2023a; Xu et al., 2023b; Li et al., 2023c; Guo et al., 2023; Kim et al., 2023a) reduce training costs by freezing quantized parameters and only training a few continuous floats. For example, PEQA (Kim et al., 2023a) and QA-LoRA (Xu et al., 2023b) focus on training continuous quantization parameters. Despite this, their performance remains poor, as depicted in Figure 1b, because the severe performance loss in low-bit scenarios (2-bit and 3-bit) cannot be fully recovered with limited trainable parameters.

To address these challenges, we introduce a novel quantization-aware training framework called EfficientQAT. This framework combines the advantages of fully trainable parameters and end-to-end training, similar to native QAT (Ma et al., 2024), while maintaining the training efficiency of PTQ (Cheng et al., 2023b; Shao et al., 2023) and Q-PEFT (Xu et al., 2023b). EfficientQAT introduces block-wise training of all parameters (Block-AP) to enhance the optimizable space and mitigate quantization accuracy loss. Block-AP sequentially trains all parameters, including original full-precision weights and quantization parameters (step sizes and zero points), within each transformer block. Several works have been developed based on block-wise reconstruction. However, previous approaches focus on designing additional trainable parameters, such as clipping thresholds for OmniQuant (Shao et al., 2023), weight rounding for AutoRound (Cheng et al., 2023b) and BRECQ (Li et al., 2021), or LoRA (Hu et al., 2021) parameters for CBQ (Ding et al., 2023). Our Block-AP is the first to directly train all parameters during block-wise reconstruction, achieving superior performance compared to previous methods (see Table 6). Block-AP successfully demonstrates that complex trainable parameter design is unnecessary for effective block-wise reconstruction in LLMs quantization. Furthermore, we introduce end-to-end training of quantization parameters (E2E-QP) to account for inter-block interactions. E2E-QP keeps the quantized weights fixed and trains only the quantization parameters (step sizes) end-to-end.

Thanks to the integration of the proposed Block-AP and E2E-QP, EfficientQAT characterizes itself as a fast-converging, memory-efficient, and high-performing quantization technique. For instance, EfficientQAT can obtain a 2-bit Llama-2-70B model on a single A100-80GB GPU in just 41 hours, with less than 3 points accuracy degradation on 5 zero-shot common-sense tasks compared to its full-precision counterpart (69.48 vs. 72.41). We also evaluate EfficientQAT across scenarios involving model compression and instruction-tuning. In model compression, as illustrated in Figure 1a, EfficientQAT significantly outperforms existing uniform quantization methods by approximately 5 points on accuracy in the challenging 2-bit quantization setting. It also matches the performance of vector quantization methods (Egiazarian et al., 2024; Tseng et al., 2024) in this scenario. In terms of instruction tuning, as shown in Figure 1b, EfficientQAT consistently outperforms existing Q-PEFT methods, including QLoRA (Dettmers et al., 2023a), QA-LoRA (Xu et al., 2023b), and PEQA (Kim et al., 2023a). For instance, EfficientQAT surpasses PEQA (Kim et al., 2023a) with 4.5 points MMLU accuracy when fine-tuning with Alpaca dataset.

## 2 RELATED WORKS

**Post-Training Quantization of LLMs.** PTQ is a pivotal technique for accelerating and deploying LLMs. Quantization approaches generally fall into two categories: weight-only quantization (Frantar et al., 2022; Dettmers et al., 2023b; Lee et al., 2023a; Kim et al., 2023b; Li et al., 2023a; Cheng et al., 2023a) and weight-activation quantization (Xiao et al., 2023; Liu et al., 2023c; Wei et al., 2022;

2023; Yuan et al., 2023; Zhao et al., 2023; Ashkboos et al., 2023; Li et al., 2023b; Ashkboos et al., 2024). Weight-only quantization focuses on compressing weights into low-bit formats, reducing memory demands and enhancing the efficiency of memory-bounded computations in LLMs (Lin et al., 2024; Yuan et al., 2024). Conversely, weight-activation quantization compresses both weights and activations, thus further decreasing the overhead associated with matrix multiplications (Lin et al., 2024). Recent advancements in weight-only quantization include the introduction of vector quantization methods by QUIP#Tseng et al. (2024) and AQLMEgiazarian et al. (2024). These methods have shown promising performance but also introduce significant overhead (Gong et al., 2024). Our research continues to explore uniform quantization, which is preferred for its compatibility with hardware implementations.

**Quantization-Aware Training of LLMs.** QAT can enhance the performance of quantized models beyond what PTQ offers. However, QAT has been less explored in LLMs due to the significant training costs involved. Studies such as LLM-QAT (Liu et al., 2023e) and BitDistiller (Du et al., 2024) investigate the application of knowledge distillation within QAT contexts. Techniques like BitNet b1.58 (Ma et al., 2024) and OneBit (Xu et al., 2024b) employ QAT to achieve extreme binary or ternary quantization levels. Although BitNet b1.58 demonstrates near-lossless performance on models up to 3 billion parameters and 100 billion training tokens with ternary quantization, its applicability to larger models or datasets remains uncertain due to prohibitive training expenses.

**Quantized Parameter-Efficient Fine-Tuning of LLMs.** Techniques like QLoRA (Dettmers et al., 2023a), INT2.1 (Chai et al., 2023), LQ-LoRA (Guo et al., 2023), and LoftQ (Li et al., 2023c) quantize model parameters to low-bit representations followed by the addition of LoRA (Hu et al., 2021) modules for fine-tuning. However, these methods require merging the LoRA modules into quantized weights, resulting in the model reverting to the FP16 format. Addressing this issue, QA-LoRA (Xu et al., 2023b) redesigns the LoRA module to merge seamlessly into the zero points. The approach most similar to ours is PEQA (Kim et al., 2023a), which uses a round-to-nearest (RTN) method for low-bit quantization and fine-tunes step sizes for task adaptation. However, PEQA experiences significant performance degradation due to limited trainable parameters, which hinders recovery from quantization information loss.

## 3 EFFICIENTQAT

### 3.1 METHOD OVERVIEW

In this section, we introduce **EfficientQAT**, a novel quantization-aware training framework for LLMs that enhances memory efficiency. As illustrated in Figure 2, traditional QAT approaches train the full-precision weights $\mathbf{W}$ and quantization parameters $s$ (step sizes) and $z$ (zero points) simultaneously in an end-to-end manner, which significantly increases the memory requirements due to the large number of parameters involved. To address this issue, EfficientQAT adopts a two-stage strategy: block-wise training of all parameters (Block-AP) and end-to-end training of quantization parameters (E2E-QP). In the Block-AP phase, model parameters and quantization parameters are trained block-by-block using reconstruction loss, which not only allows for precise calibration with full training but also reduces memory consumption (Li et al., 2021; Shao et al., 2023) by block-wise training. Following this, the E2E-QP phase fixes the quantized weights and trains the step sizes exclusively on target datasets, thus achieving inter-block interaction in a memory-efficient way. Details on Block-AP and E2E-QP are further described in Sections 3.2 and 3.3, respectively.

### 3.2 BLOCK-WISE TRAINING OF ALL PARAMETERS

In this section, we introduce the Block-Wise Training of All Parameters (Block-AP) approach, designed to efficiently provide an effective initialization for following end-to-end training.

**Quantization and Dequantization.** Specifically, Block-AP begins with a standard uniform quantization method:

$$\mathbf{W}_{int} = \text{clamp}(\lfloor \frac{\mathbf{W}}{s} \rceil + z, 0, 2^N - 1), \tag{1}$$

where $\lfloor \cdot \rceil$ represents the rounding operation. $N$ is the target bit number. $\mathbf{W}_{int}$ and $\mathbf{W}$ denote the quantized integer and full-precision weights, respectively. $s$ is the scaling factor and $z$ is the zero

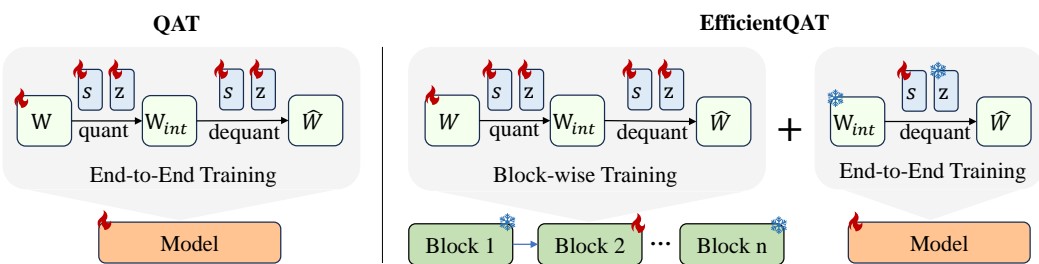

Figure 2: **The overall pipeline of naive QAT and proposed EfficientQAT.** EfficientQAT introduces two novel processes: Block-wise Training of All Parameters (Block-AP) and End-to-End Training of Quantization Parameters (E2E-QP).

point. In the forward propagation, the quantized weights are converted back to full precision as follows:

$$\widehat{\mathbf{W}} = (\mathbf{W_{int}} - z) \cdot s. \tag{2}$$

Here, $\widehat{\mathbf{W}}$ refers to the dequantized weights used in the forward computation. The processes of quantization (Eq.(1)) and dequantization (Eq.(2)) are integrated within the computation graph and can be optimized through gradient descent in a quantization-aware manner.

**Blcok-wise Quantization-aware Training.** Traditional QAT methods (Ma et al., 2024; Esser et al., 2019; Liu et al., 2023e) train the entire network using Eq.(1) and Eq.(2) in an end-to-end fashion, which typically requires substantial computational resources and extensive data to prevent overfitting. Here we aim to enhance the training efficiency of QAT. Previous studies, such as BRECQ (Li et al., 2021), have demonstrated that block-wise training achieves faster convergence and requires less training time, data, and memory than end-to-end training given a pre-trained model. Following the methodologies in BRECQ (Li et al., 2021) and OmniQuant (Shao et al., 2023), Block-AP sequentially conducts quantization-aware training within one transformer block before moving on to the next under a block-wise reconstruction framework.

**Full Training of Model Weights and Quantization Parameters.** Unlike previous methods which optimize several quantization parameters such as rounding parameters (Nagel et al., 2020; Cheng et al., 2023b; Lee et al., 2023b), clipping parameters (Shao et al., 2023), and step sizes (Esser et al., 2019; Ding et al., 2023), Block-AP behaves like QAT, training all inherent parameters from Eq.(1) and Eq.(2), including scaling factor $s$, zero point $z$, and model weights $\mathbf{W}$.

In our Block-AP approach, a straightforward full-training regimen outperforms existing partial-training variants (Nagel et al., 2020; Li et al., 2021; Ding et al., 2023) with intricate designs. Traditional training methods involving rounding parameters (Nagel et al., 2020; Li et al., 2021; Ding et al., 2023) serve as regularization techniques, constraining the update range of integral weights to $(-1, +1)$ to mitigate overfitting. However, this approach limits the solution space, potentially hindering the final performance of quantized models. Our empirical findings demonstrate the superiority of full training within our Block-AP over existing partial-training variants (Nagel et al., 2020; Li et al., 2021; Ding et al., 2023), as shown in Table 6.

Following block-wise training, we obtain the quantized model which includes quantized weights $\mathbf{W}_q$, step sizes $s$, and zero points $z$ for each quantization group. The weights $\mathbf{W}_q$ and zero points $z$ are stored in a low-bit format, while step sizes $s$ are stored in FP16. Note that $s$ and $z$ are shared within their respective quantization groups and constitute only a small fraction of the model's parameters, approximately 1.6% for a group size of 64. Moreover, the model's memory footprint is substantially reduced by transitioning from full-precision 16-bit weights to 2/3/4-bit quantized weights.

### 3.3 END-TO-END TRAINING OF QUANTIZATION PARAMETERS

We further introduce the End-to-End Training of Quantization Parameters (E2E-QP), aimed at efficiently training the entire quantized model on target datasets.

**End-to-End Training of step sizes.** Unlike traditional Quantization-Aware Training (QAT) methods (Liu et al., 2023e; Ma et al., 2024) that train full-precision weights, E2E-QP begins with $\mathbf{W}_q$

initialized via Block-AP and focuses solely on the training of quantization parameters ($s$ and $z$). Our findings indicate that training $s$, $z$, or both yields similar performance (see Table 7 for details). However, since training $z$ involves converting it from a low-bits format to full-precision, we typically train only $s$ by default unless specified otherwise to avoid additional memory overhead.

Additionally, within E2E-QP, there is no quantization process as per Equation (1); only the dequantization process occurs as described in Equation (2). Thus, the gradient of the trainable parameter $s$ is computed as $\frac{\partial \widehat{w}}{\partial s} = w_q - z$.

Overall, the memory usage for training in E2E-QP is drastically reduced due to the reduced trainable parameter count. Detailed memory footprints for various model sizes and bits under E2E-QP are listed in Table 8. For instance, the Llama-2-70B model can complete 2-bit QAT through E2E-QP using only 34.2GB of memory. Equipped with E2E-QP, EfficientQAT is adaptable to different scenarios by simply changing the training datasets, which includes applications such as continual pre-training and instruction-tuning (Taori et al., 2023).

Table 1: Llama 2 & 3 average zero-shot accuracy on 5 common-sense reasoning tasks (↑). Detailed number of each task can be found at Table 15 (4-bit), Table 16 (3-bit), and Table 17 (2-bit). "Group" indicates group size for uniform quantization and codebook scheme for vector quantization. "**bold**" indicates best results for uniform quantization.

| Method | Bits | Type | Group (code) | 2-7 | 2-13 | 2-70 | 3-8 | 3-70 |
|---|---|---|---|---|---|---|---|---|
| FP16 | 16 | - | 16 | 64.86 | 67.81 | 72.41 | 68.58 | 75.33 |
| RTN | 4 | uniform | 128 | 64.52 | 67.50 | 72.26 | 67.79 | 73.98 |
| GPTQ | 4 | uniform | 128 | 64.24 | 67.27 | 72.39 | 67.80 | 74.74 |
| AWQ | 4 | uniform | 128 | **64.54** | **67.61** | 72.44 | 68.24 | **74.77** |
| OmniQ | 4 | uniform | 128 | 64.52 | 67.10 | 72.39 | - | - |
| AutoRound | 4 | uniform | 128 | 64.39 | 67.36 | 72.47 | - | - |
| QuIP# | 4 | vector | - | 64.48 | 67.28 | 72.17 | - | - |
| EfficientQAT | 4 | uniform | 128 | 64.27 | 67.52 | **72.62** | **68.43** | 74.57 |
| RTN | 3 | uniform | 128 | 62.06 | 65.77 | 70.83 | 58.72 | 65.29 |
| GPTQ | 3 | uniform | 128 | 62.48 | 66.18 | 71.47 | 60.58 | 71.28 |
| AWQ | 3 | uniform | 128 | 62.82 | 66.14 | 71.41 | 64.82 | 73.65 |
| OmniQ | 3 | uniform | 128 | 62.42 | 66.18 | 71.07 | - | - |
| AutoRound | 3 | uniform | 128 | 63.72 | 66.68 | 71.24 | - | - |
| QuIP# | 3 | vector | - | 63.52 | 66.26 | 72.13 | - | - |
| EfficientQAT | 3 | uniform | 128 | **64.02** | **67.28** | 71.76 | **67.35** | **72.42** |
| OmniQ | 2 | uniform | 128 | 46.98 | 53.56 | 54.87 | - | - |
| AutoRound | 2 | uniform | 128 | 54.50 | 60.72 | 67.70 | - | - |
| EfficientQAT | 2 | uniform | 128 | **59.50** | **63.88** | 68.93 | 59.37 | 67.57 |
| AQLM | 2 | vector | 2x8 | 57.61 | 62.22 | 69.85 | - | - |
| AQLM | 2 | vector | 1x16 | 61.85 | 64.95 | 70.84 | 64.10 | 70.10 |
| QuIP# | 2 | vector | - | 60.61 | 64.44 | 70.91 | - | - |
| EfficientQAT | 2 | uniform | 64 | **60.14** | **64.48** | **69.48** | **60.76** | **67.89** |

## 4 EXPERIMENTS

This section presents extensive experiments to verify our proposed EfficientQAT. Secition 4.1 and Sec 4.2 present the comparisons with quantization methods and Q-PEFT methods respectively. Section 4.4 details the training cost and inference speed-up of the proposed EfficientQAT. Section 4.3 presents the comprehensive ablation studies of the proposed EfficientQAT.

### 4.1 EFFICIENTQAT FOR LLMS QUANTIZATION

**Training.** We conduct experiments on the Llama-2 and Llama-3 models. For Block-AP, we use 4096 samples from RedPajama (Computer, 2023) with a context length of 2048. We train each block with batch size as 2 and epochs as 2, setting the learning rate of quantization parameters as $1 \times 10^{-4}$, and the learning rate of weights as $2 \times 10^{-5}$ for 2-bit and $1 \times 10^{-5}$ for 3/4-bits. For E2E-QP, we also

employ 4096 samples from RedPajama (Computer, 2023) but with a context length of 4096. We train the entire model with batch size as 32 and epoch as 1, and set the learning rate of step size as $2 \times 10^{-5}$ for 2-bit and $1 \times 10^{-5}$ for 3/4-bits.

**Evaluation.** We assess the zero-shot accuracy of five common-sense reasoning tasks using the v0.4.2 lm-evaluation-harness[1]. The tasks include WinoGrande (Sakaguchi et al., 2021), PIQA (Bisk et al., 2020), HellaSwag (Zellers et al., 2019), Arc-Easy (Clark et al., 2018), and Arc-Challenge (Clark et al., 2018). We also measure the perplexity of Wikitext2 and C4 with a 2048 context length, as done in previous studies (Frantar et al., 2022; Shao et al., 2023).

**PTQ Baseline.** We compare our results with PTQ methods from uniform quantization such as GPTQ (Frantar et al., 2022), AWQ (Lin et al., 2023), OmniQ (Shao et al., 2023), ApiQ (Liao & Monz, 2024) and AutoRound (Cheng et al., 2023b), and vector quantization including QuIP# (Tseng et al., 2024) and AQLM (Egiazarian et al., 2024). Note that if a result is the best of uniform quantization, we set it to **bold**.

Table 2: Comparison with QAT methods on Llama-2 & 3 models.

| Model | Method | Bits | Group | PPL | | Average |
|---|---|---|---|---|---|---|
| | | | | Wiki | C4 | Accuracy |
| 2-7B | LLM-QAT | 3 | 128 | 6.02 | - | - |
| 2-7B | BitDistiller | 3 | 128 | 5.97 | - | - |
| 2-7B | EfficientQAT | 3 | 128 | **5.81** | **7.34** | **64.02** |
| 2-7B | PB-LLN | 2 | 64 | 20.37 | 44.88 | 37.59 |
| 2-7B | DB-LLN | 2 | 64 | 7.23 | 9.62 | 55.12 |
| 2-7B | EfficientQAT | 2 | 64 | 6.86 | 8.50 | 60.14 |
| 2-7B | LLM-QAT | 2 | 128 | 9.30 | - | - |
| 2-7B | BitDistiller | 2 | 128 | 8.08 | - | - |
| 2-7B | EfficientQAT | 2 | 128 | **7.19** | **8.79** | **59.50** |
| 3-8B | PB-LLM | 2 | 128 | 24.70 | 79.20 | 38.80 |
| 3-8B | DB-LLM | 2 | 128 | 13.60 | 19.20 | 51.80 |
| 3-8B | EfficientQAT | 2 | 128 | **9.80** | **13.22** | **59.37** |
| 3-70B | PB-LLM | 2 | 128 | 11.60 | 34.50 | 46.00 |
| 3-70B | EfficientQAT | 2 | 128 | **6.07** | **9.23** | **67.57** |

**PTQ Results.** The accuracy and perplexity results are presented in Table 1 and Table 3, respectively. It seems that minimal performance degradation with 4-bit group quantization, and even simple round-to-nearest (RTN) quantization achieves respectable results. Therefore, EfficientQAT offers a slight perplexity improvement (∼0.02) in 4-bit uniform quantization and comparable zero-shot accuracy with existing methods. As the number of quantization bits decreases, EfficientQAT's performance advantage increases. In 3-bit quantization, EfficientQAT surpasses the optimization-based method AutoRound by approximately 0.5% in zero-shot accuracy and outperforms OmniQuant by 0.14 to 0.43 in perplexity. Notably, in 2-bit quantization, EfficientQAT outperforms AutoRound (Cheng et al., 2023b) by 5% in zero-shot accuracy for Llama-2-7b. EfficientQAT also exceeds previous vector quantization methods like AQLM (Egiazarian et al., 2024) using a 2x8 codebook. While AQLM (Egiazarian et al., 2024) with a 1x16 codebook and QuIP# (Tseng et al., 2024) perform slightly better in 2-bit scenarios, they introduce notable computational overhead (Gong et al., 2024) and can even reduce inference speed. Thus, EfficientQAT successfully pushes the limits of uniform quantization in LLMs, making hardware-friendly uniform quantization competitive with more complex vector quantization methods. However, we can also find that Llama-3 models suffer more performance degeneration after quantization (Huang et al., 2024), which may caused by the full training with 15T tokens other than 2T tokens.

**QAT Baseline.** We also compare our results with existing QAT methods, including LLM-QAT (Liu et al., 2023e), BitDistiller (Du et al., 2024), PB-LLM (Shang et al., 2023) and DB-LLM (Chen et al., 2024).

**QAT Results.** As demonstrated in Table 2, EfficientQAT markedly surpasses other QAT methods in performance. Specifically, EfficientQAT reduces perplexity by 0.89 compared to BitDistiller (Du et al., 2024) when applied to the Llama-2-7B model with w2g128 quantization. In the more challenging scenario of Llama-3 quantization (Huang et al., 2024), EfficientQAT exceeds the performance of DB-LLM (Chen et al., 2024) by 5.98 in C4 perplexity and by 8.57 points in average accuracy with the w2g128 quantization.

---

[1]https://github.com/EleutherAI/lm-evaluation-harness

Table 3: Llama 2 & 3 Wikitext2 and C4 perplexity (↓), context length 2048.

| Method | Bits | Type | Group | Wikitext 2 | | | | | C4 | | | | |
|---|---|---|---|---|---|---|---|---|---|---|---|---|---|
| | | | | 2-7 | 2-13 | 2-70 | 3-8 | 3-70 | 2-7 | 2-13 | 2-70 | 3-8 | 3-70 |
| FP16 | 16 | - | 16 | 5.47 | 4.88 | 3.32 | 6.14 | 2.85 | 6.97 | 6.47 | 5.52 | 8.88 | 6.73 |
| GPTQ | 4 | uniform | 128 | 5.61 | 4.98 | 3.42 | 6.53 | 3.38 | 7.13 | 6.56 | 5.58 | 9.35 | 7.02 |
| AWQ | 4 | uniform | 128 | 5.62 | 4.97 | 3.41 | 6.55 | 3.26 | 7.13 | 6.56 | 5.58 | 9.41 | 6.98 |
| OmniQ | 4 | uniform | 128 | 5.58 | 4.95 | 3.40 | - | - | 7.12 | 6.56 | 5.58 | - | - |
| QuIP# | 4 | vector | - | 5.56 | 4.95 | 3.38 | - | - | 7.07 | 6.54 | 5.56 | - | - |
| EfficientQAT | 4 | uniform | 128 | **5.53** | **4.93** | **3.39** | **6.47** | **3.17** | **7.07** | **6.54** | **5.58** | **9.26** | **6.94** |
| GPTQ | 3 | uniform | 128 | 6.29 | 5.42 | 3.85 | 9.58 | 5.25 | 7.89 | 7.00 | 5.85 | 11.66 | 8.64 |
| AWQ | 3 | uniform | 128 | 6.24 | 5.32 | 3.74 | 8.16 | 4.69 | 7.84 | 6.94 | 5.81 | 11.49 | 7.91 |
| OmniQ | 3 | uniform | 128 | 6.03 | 5.28 | 3.78 | - | - | 7.75 | 6.98 | 5.85 | - | - |
| QuIP# | 3 | vector | - | 5.79 | 5.10 | 3.56 | - | - | 7.32 | 6.72 | 5.67 | - | - |
| EfficientQAT | 3 | uniform | 128 | **5.81** | **5.12** | **3.61** | **7.09** | **4.19** | **7.34** | **6.73** | **5.71** | **10.06** | **7.43** |
| OmniQ | 2 | uniform | 128 | 11.06 | 8.26 | 6.55 | - | - | 15.02 | 11.05 | 8.52 | - | - |
| ApiQ | 2 | uniform | 128 | 8.25 | 6.71 | - | - | - | 12.04 | 9.13 | - | - | - |
| EfficientQAT | 2 | uniform | 128 | **7.19** | **6.08** | **4.61** | **9.80** | **6.38** | **8.79** | **7.75** | **6.48** | **13.22** | **9.53** |
| AQLM | 2 | vector | 2x8 | 7.24 | 6.06 | 4.49 | - | - | 8.96 | 7.80 | 6.36 | - | - |
| AQLM | 2 | vector | 1x16 | 6.34 | 5.59 | 4.06 | 7.76 | 5.10 | 8.06 | 7.27 | 6.04 | 10.89 | 8.31 |
| QuIP# | 2 | vector | - | 6.66 | 5.74 | 4.16 | - | - | 8.35 | 7.45 | 6.12 | - | - |
| OmniQ | 2 | uniform | 64 | 9.62 | 7.56 | 6.11 | - | - | 12.72 | 10.05 | 7.68 | - | - |
| ApiQ | 2 | uniform | 64 | 7.59 | 6.44 | - | - | - | 10.56 | 8.92 | - | - | - |
| EfficientQAT | 2 | uniform | 64 | **6.86** | **5.96** | **4.52** | **9.41** | **6.07** | **8.50** | **7.59** | **6.38** | **12.77** | **9.23** |

## 4.2 EFFICIENTQAT FOR INSTRUCTION TUNING

**Training and Evaluation.** Following existing works (Xu et al., 2023b; Qin et al., 2024), we train Llama-1 models on the Alpaca dataset (Taori et al., 2023) and assess their performance by measuring average 5-shot MMLU (Hendrycks et al., 2020) accuracy works (Xu et al., 2023b; Qin et al., 2024). The training hyperparameters are identical to those described in Section 4.1, except we replace the RedPajama dataset (Computer, 2023) with Alpaca. In line with QLoRA's methodology (Dettmers et al., 2023a), we adjust the source context length to 384 and the target context length to 128, training for 10,000 steps with a batch size of 16.

**Baseline.** We benchmark EfficientQAT against several leading methods, including QLoRA (Dettmers et al., 2023a), QA-LoRA (Xu et al., 2023b), PEQA (Kim et al., 2023a), and IR-QLoRA (Qin et al., 2024), across quantization setting of 2, 3, and 4 bits. Consistent with QA-LoRA (Xu et al., 2023b), we also employ GPTQ (Frantar et al., 2022) to quantize the fine-tuned QLoRA models into a low-bit format without FP16 LoRA for equitable comparison.

**Results.** Both Table 4 and Figure 1b indicate that EfficientQAT significantly outperforms existing Q-PEFT methods. For instance, in channel-wise quantization (group size of -1), EfficientQAT achieves more than 3% higher accuracy than PEQA (Kim et al., 2023a). In the 2-bit quantization scenario, the superiority of EfficientQAT is even more pronounced, surpassing QA-LoRA (Xu et al., 2023b) by 5.1% and 4.0% in 7B and 13B models, respectively, and outperforming PEQA by 4.5% and 8.7% in the same models. Moreover, Table 4 also demonstrates that EfficientQAT outperforms both QA-LoRA and QLoRA with GPTQ in smaller model memory footprint (larger group size).

## 4.3 ABLATION ANALYSIS

The EfficientQAT algorithm is comprised of two main components: Block-AP and E2E-QP. This section evaluates the effectiveness, trainable parameters, and training sample requirements of each component. We present the average perplexity for WikiText2 and C4 datasets, and the average accuracy for five zero-shot reasoning tasks, similar to Table 1.

**Effectiveness of each component.** As indicated in Table 5, both the Block-AP and E2E-QP components significantly enhance performance, with their combination yielding the best results. Notably, Block-AP outperforms E2E-QP, aligning with findings from BRECQ (Li et al., 2021).

Table 4: Llama-1 average MMLU accuracy (5-shot) about instruction-tuning on Alpaca dataset.

| Method | Bits | Group | 7B | 13B |
|---|---|---|---|---|
| - | 16 | - | 34.6 | 46.3 |
| PEQA | 4 | -1 | 35.8 | 45.0 |
| EfficientQAT | 4 | -1 | **38.8** | **48.2** |
| QLoRA | 4+16 | - | 38.4 | 48.4 |
| QLoRA w/GPTQ | 4 | 32 | 36.0 | 48.0 |
| QA-LoRA | 4 | 32 | 39.4 | 49.2 |
| PEQA | 4 | 64 | 39.4 | 47.4 |
| IR-QLoRA | 4 | 64 | 40.8 | 49.3 |
| EfficientQAT | 4 | 64 | **41.2** | **49.5** |
| QLoRA w/ GPTQ | 3 | 32 | 34.0 | 46.1 |
| QA-LoRA | 3 | 32 | 37.4 | 47.3 |
| IR-QLoRA | 3 | 64 | 38.4 | - |
| PEQA | 3 | 64 | 38.5 | 46.3 |
| EfficientQAT | 3 | 64 | **40.0** | **48.2** |
| QLoRA w/ GPTQ | 2 | 32 | 25.8 | 30.9 |
| QA-LoRA | 2 | 32 | 27.5 | 36.9 |
| IR-QLoRA | 2 | 64 | 27.8 | - |
| PEQA | 2 | 64 | 28.1 | 32.2 |
| EfficientQAT | 2 | 64 | **32.6** | **40.9** |

Table 5: Effectiveness of each component on Llama-2-7B w2g64 quantization.

| Block-AP | E2E-QP | Avg. PPL | Avg. Accuracy |
|---|---|---|---|
| ✗ | ✗ | 453.49 | 40.69 |
| ✓ | ✗ | 8.53 | 58.99 |
| ✗ | ✓ | 9.33 | 55.71 |
| ✓ | ✓ | 7.68 | 60.14 |

Table 6: W2g64 Llama-2-7B performance with different trainable parameters in the block-wise training (w/o E2E-QP). "# Param." indicates trainable parameters count in a block.

| Param. | # Param. | Memory | Avg. PPL | Avg. Accuracy |
|---|---|---|---|---|
| clipping | 6.3M | 6.4GB | 11.28 | 53.20 |
| $s,z$ | 6.3M | 6.4GB | 10.26 | 55.20 |
| round | 202.4M | 8.6GB | 15.50 | 45.32 |
| $s,z$,round | 208.7M | 9.3GB | 9.17 | 57.14 |
| $s,z$,**W** | 208.7M | 8.5GB | 8.53 | 58.99 |

Table 7: Llama-2-7B w2g64 quantization with different trainable parameters for E2E-QP (w/ Block-AP).

| Param. | Avg. Bits | Avg. PPL | Avg. Accuracy |
|---|---|---|---|
| $s$ | 2.28 | 7.68 | 60.14 |
| $z$ | 2.50 | 7.69 | 60.08 |
| $s, z$ | 2.50 | 7.68 | 60.18 |

**Trainable parameters of Block-AP.** Block-AP trains all parameters, including original weights and quantization parameters. Previous methods have introduced various training strategies to mitigate overfitting, such as trained rounding (Nagel et al., 2020; Cheng et al., 2023b), clipping thresholds (Shao et al., 2023), and step sizes (Esser et al., 2019; Ding et al., 2023). We compare Block-AP with these methods by modifying only the trainable parameters of Block-AP. As shown in Table 6, Block-AP (training $s$, $z$, **W**) performs best with an acceptable training cost. Additionally, the memory footprint of directly training **W** is even smaller than that of training the rounding operation, which requires an additional copy of rounding parameters.

**Trainable parameters of E2E-QP.** We further examine the trainable parameters within E2E-QP. Table 7 shows that training $s$, $z$, or both yields similar performance. However, given that converting $z$ from an original low-bit representation to a trainable FP16 format increases the average bit count, we opt to train only $s$ by default.

**Samples number of Block-AP.** We assess the number of training samples for Block-AP, noting that E2E-QP trains all parameters, which may lead to overfitting. To address this, we introduce an additional 64 unseen samples from ReadPajama to evaluate the overfitting issue. We adjust the training epochs to ensure a similar total training time, allowing for fair comparisons across different sample sizes. As illustrated in Figure 3, increasing the number of training samples significantly reduces the gap between training loss and validation loss from 1.07 to 0.06. This reduction corresponds to an increase in the average accuracy for zero-shot tasks from 57.14% to 58.99%. Consequently, we set the default number of training samples for E2E-QP at 4096, as this maintains a minimal gap between training and validation losses.

**Samples number of E2E-QP.** In the E2E-QP, we train the model for 1 epoch to avoid over-fitting. Our examination of the training sample sizes for E2E-QP, detailed in Table 4, reveals that average perplexity consistently improves as sample sizes increase from 128 to 32,674. However, there is no significant improvement in average accuracy beyond 4096 samples. Therefore, we set the training sample size for E2E-QP at 4096 by default to balance efficiency and performance. Nonetheless, it is possible to further enhance the performance of EfficientQAT by increasing the sample size.

### 4.4 EFFICIENCY OF EFFICIENTQAT

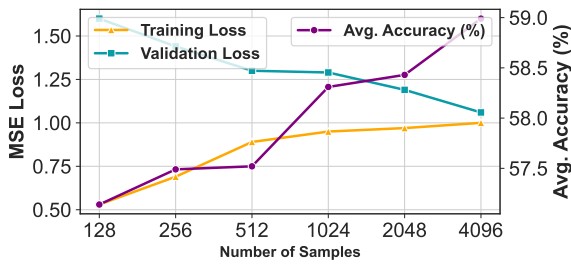

Figure 4: Llama-2-7B w2g64 quantization performance with different sample numbers for E2E-QP (w/ Block-AP).

| # Samples | Avg. PPL | Avg. Accuracy |
|---|---|---|
| 128 | 8.09 | 59.03 |
| 512 | 7.88 | 59.81 |
| 2048 | 7.75 | 60.13 |
| 4096 | 7.68 | 60.14 |
| 8192 | 7.63 | 60.19 |
| 32764 | **7.50** | **60.31** |

Figure 3: Illustration of training loss, validation loss and average accuracy of w2g64 Llama-2-7b with different training samples size for Block-AP (w/o E2E-QP).

Table 8: The detailed training time and training memory of EfficientQAT across different model size and quantization bits on a single A100-80GB GPU.

| Llama-2 | Block-AP | | E2E-QP | | |
|---|---|---|---|---|---|
| | Time | Memory | Time | Memory (4-/3-/2-bits) | Total Time |
| 7B | 3.3h | 8.5GB | ~1.5h | 7.0/6.4/5.6GB | 4.8h |
| 13B | 5.6h | 10.3GB | ~2.9h | 11.7/10.6/9.1GB | 8.5h |
| 70B | 26.6h | 29.9GB | ~14.3h | 48.4/42.0/34.2GB | 40.9h |

**Training Efficiency** Table 8 illustrates the required memory and time for training Lllama-2 models using EfficientQAT. The results indicate that the model completes training rapidly, taking 4.8 hours for the 7B model and 40.9 hours for the 70B model. we further compare the training time with other QAT methods, including Bit-Distiller, PB-LLM, and DN-LLM. As shown in Table 1c, the training time of EfficientQAT is significantly lower than that of existing methods. For example, the tuning time of EfficientQAT is only 14% of AQLM and 50% of DB-LLM. Additionally, for quantizing a 70B model, the full process of EfficientQAT can be completed on a single A100-80GB GPU. However, other

Table 9: Comparisons of training time with existing methods in Llama-2-70B.

| Method | One A100-80GB? | GPU hours (h) |
|---|---|---|
| LLM-QAT | ✗ | 900 |
| QuIP# | ✗ | 300 |
| AQLM | ✓ | 336 |
| BitDistiller | ✗ | 64 |
| PB-LLM | ✗ | 90 |
| DB-LLM | ✗ | 82 |
| **EfficientQAT** | ✓ | **41** |

methods, except AQLM, require at least 4 A100-80GB GPUs to quantize a model of this size. Therefore, EfficientQAT is both a time-efficient and memory-efficient QAT method.

**Inference Efficiency** Due to the leverage of standard uniform quantization, the quantized models of EfficientQAT can also achieve speedup through a lot of toolboxes, such as MLC-LLM (team, 2023), AWQ (Lin et al., 2023), and BitBLAS (Wang et al., 2024), T-MAC (Wei et al., 2024), Marlin (Frantar et al., 2024), *etc*. For example, Table 10 shows that INT2 quantization of EfficientQAT can enhance the forward-pass speed by approximately 2.9x to 4.4x through BitBLAS (Wang et al., 2024).

## 5 CONCLUSION

In this study, we introduce EfficientQAT, a novel method that completes QAT with improved efficiency in both memory usage and training time. Through comprehensive testing, EfficientQAT proves superior to existing PTQ, QAT, and Q-PEFT methods in terms of versatility and performance across various models and quantization levels. Additionally, EfficientQAT leverages a standard uniform quantization, which simplifies deployment using popular toolboxes. We anticipate that EfficientQAT will stimulate further research and improve the compression of Large Language Models (LLMs), making them more efficient and widely accessible.

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

- Sec. H presents the detailed accuracy for each zero-shot task.

## A    REPRODUCIBILITY STATEMENT

In this section, we summarize the necessary information to reproduce our results. We provide the training and evaluation details at the beginning of each sub-section in Sec. 4. We also provide the source of detailed results for each compared method in Sec.D.

## B    GRADIENT OF TRAINABLE PARAMETERS IN BLOCK-AP

Block-AP, aligned with LSQ+(Bhalgat et al., 2020), uses a straight-through estimator (STE)(Bengio et al., 2013) to facilitate gradient computation through the rounding operation. The gradients of scaling factor $s$ are computed as follows:

$$\frac{\partial \widehat{w}}{\partial s} = \begin{cases} \lfloor \frac{w}{s} \rceil - \frac{w}{s}, 0 \leq \lfloor \frac{w}{s} \rceil + z \leq 2^{N-1}, \\ -z, \lfloor \frac{w}{s} \rceil + z < 0, \\ 2^{N-1} - z, \lfloor \frac{w}{s} \rceil + z > 2^{N-1}. \end{cases} \tag{3}$$

and the gradient with respect to zero point $z$ is:

$$\frac{\partial \widehat{w}}{\partial z} = \begin{cases} 0, 0 \leq \lfloor \frac{w}{s} \rceil + z \leq 2^{N-1}, \\ -1, otherwise, \end{cases} \tag{4}$$

and the full-precision weight $\mathbf{W}$ can also be updated through its gradient[2]:

$$\frac{\partial \widehat{w}}{\partial w} = \begin{cases} 1, 0 \leq \lfloor \frac{w}{s} \rceil + z \leq 2^{N-1}, \\ 0, otherwise, \end{cases} \tag{5}$$

## C    SPEEDUP WITH BITBLAS

According to Table 10, INT2 quantization enhances the forward-pass speed by approximately 2.9x to 4.4x.

---

[2] $\widehat{w},w$ is a element from $\widehat{W}$, $\mathbf{W}$

Table 10: Speed of the FP16 linear layer matrix-vector multiplication in PyTorch, and relative INT2 speedups in BitBLAS Wang et al. (2024). Testing on A100-80GB GPU.

| Llama-2 | 7B | | 13B | | 70B | |
|---|---|---|---|---|---|---|
| size (out_c $\times$ in_c) | 4096x4096 | 11008x4096 | 5120x5120 | 13824x5120 | 8192x8192 | 28672x8192 |
| FP16 | 25 us | 61 us | 38 us | 90 us | 91 us | 286 us |
| INT2 | 9 us | 21 us | 11 us | 26 us | 24 us | 67 us |
| Speedup | 3.1x | 2.9x | 3.6x | 3.5x | 3.9x | 4.4x |

## D  RESULTS SOURCE OF OTHER METHOD.

In this study, we present a thorough comparison of our method against existing PTQ techniques, including GPTQ (Frantar et al., 2022), AWQ (Lin et al., 2023), OmniQ (Shao et al., 2023), AutoRound (Cheng et al., 2023b), QuIP# (Tseng et al., 2024), and AQLM (Egiazarian et al., 2024). We also compare with existing QAT methods, including LLM-QAT (Liu et al., 2023e), BitDistiller (Du et al., 2024), PB-LLM (Shang et al., 2023) and DB-LLM (Chen et al., 2024). Additionally, we also evaluate quantized parameter-efficient fine-tuning methods such as PEQA (Kim et al., 2023a), QLoRA (Dettmers et al., 2023a), QA-LoRA (Xu et al., 2023b), and IR-QLoRA (Qin et al., 2024). The results we discuss originate from their respective official publications, and other scholarly articles, or are derived from our reproduction. We meticulously document the source of the results for each method as follows:

- GPTQ, AWQ, OmniQ, AutoRound: The zero-shot accuracy results for Llama-2 models using these methods are derived from the AutoRound GitHub repository[3]. The perplexity results for the Llama-2 models using GPTQ, AWQ, and OmniQ are taken from the OmniQ paper (Shao et al., 2023). The results for Llama-3 models using AWQ[4] and GPTQ[5] were obtained through their open-source implementations.

- QuIP#, AQLM: We replicated the results using the official pre-trained models provided by QuIP#[6] and AQLM[7].

- LLM-QAT, BitDistiller: These results are cited from BitDistiller (Du et al., 2024) paper.

- PB-LLM, DB-LLM: These results are cited from recent Llama-3 quantization empirical study (Huang et al., 2024).

- ApiQ: These results are cited from IR-ApiQ (Liao & Monz, 2024) paper.

- PEQA: The per-channel quantization results (g=-1) are cited from their publication (Kim et al., 2023a), and the results for a group size of 64 were produced using our codebase.

- QA-LoRA, QLoRA, QLoRA w/ GPTQ: These results are cited from QA-LoRA (Xu et al., 2023b) paper.

- IR-QLoRA: These results are cited from IR-QLoRA (Qin et al., 2024) paper.

## E  SIZE OF QUANTIZED MODELS

This section illustrates model size reduction achieved through quantization. Models quantized to low-bit representations are more compact.

We implement N-bit quantization with a grouping size of $g$, where each group of $g$ weights shares the same FP16 step size and an N-bit zero point. Consequently, the average number of bits per parameter is calculated as $N + \frac{N+16}{g}$. It is important to note that only the linear layers within the transformer

---

[3]AutoRound: https://github.com/intel/auto-round/blob/main/docs/acc.md

[4]AWQ:https://github.com/mit-han-lab/llm-awq

[5]GPTQ:https://github.com/qwopqwop200/GPTQ-for-LLaMa

[6]https://github.com/Cornell-RelaxML/quip-sharp

[7]https://github.com/Vahe1994/AQLM

Table 11: **Model size of quantized models.** Compression ratio indicates the compression ratio of quantized models compared with FP16 models.

| Model | # Bit | Group size | bits/param | size (GiB) | Compression ratio (%) |
|---|---|---|---|---|---|
| LLaMA-2-7B | 16 | - | 16 | 12.55 | - |
| | 4 | 32 | 4.63 | 3.98 | 68.33 |
| | 4 | 64 | 4.31 | 3.74 | 70.20 |
| | 4 | 128 | 4.16 | 3.62 | 71.14 |
| | 3 | 32 | 3.59 | 3.35 | 73.28 |
| | 3 | 64 | 3.30 | 3.13 | 75.08 |
| | 3 | 128 | 3.15 | 3.01 | 75.98 |
| | 2 | 32 | 2.56 | 2.42 | 80.71 |
| | 2 | 64 | 2.28 | 2.21 | 82.40 |
| | 2 | 128 | 2.14 | 2.10 | 83.25 |
| LLaMA-2-13B | 16 | - | 16 | 24.24 | - |
| | 4 | 32 | 4.63 | 7.44 | 69.30 |
| | 4 | 64 | 4.31 | 6.98 | 71.21 |
| | 4 | 128 | 4.16 | 6.75 | 72.16 |
| | 3 | 32 | 3.59 | 6.22 | 74.33 |
| | 3 | 64 | 3.30 | 5.78 | 76.16 |
| | 3 | 128 | 3.15 | 5.56 | 77.07 |
| | 2 | 32 | 2.56 | 4.40 | 81.87 |
| | 2 | 64 | 2.28 | 3.98 | 83.58 |
| | 2 | 128 | 2.14 | 3.77 | 84.44 |
| LLaMA-2-70B | 16 | - | 16 | 128.48 | - |
| | 4 | 32 | 4.63 | 37.83 | 70.55 |
| | 4 | 64 | 4.31 | 35.34 | 72.49 |
| | 4 | 128 | 4.16 | 34.10 | 73.46 |
| | 3 | 32 | 3.59 | 31.26 | 75.67 |
| | 3 | 64 | 3.30 | 28.87 | 77.53 |
| | 3 | 128 | 3.15 | 27.67 | 78.46 |
| | 2 | 32 | 2.56 | 21.40 | 83.34 |
| | 2 | 64 | 2.28 | 19.16 | 85.09 |
| | 2 | 128 | 2.14 | 18.04 | 85.96 |

blocks are quantized; other layers, such as normalization layers, embeddings, and the classification head, remain in FP16 format. Table 11 provides detailed comparisons of quantized model sizes and their compression ratios.

Table 12: Lllma-2-7B 2-bit quantization performance with different group sizes for proposed EfficientQAT.

| Group | Avg. Bits | Avg. PPL | Avg. Accuracy |
|---|---|---|---|
| 32 | 2.56 | 7.59 | 60.28 |
| 64 | 2.28 | 7.68 | 60.14 |
| 128 | 2.10 | 7.99 | 59.50 |
| 256 | 2.07 | 8.18 | 58.67 |

Table 13: Block-AP (w/o E2E-QP) results of Llama-2-7B in different calibration datasets.

| Bits | Dataset | Wiki PPL | C4 PPL | Avg. Accuracy |
|---|---|---|---|---|
| w3g128 | WikiText2 | 5.72 | 7.52 | 63.24 |
| w3g128 | C4 | 5.92 | 7.38 | 63.82 |
| w3g128 | Redpajama | 5.91 | 7.41 | 63.50 |
| w2g64 | WikiText2 | 6.73 | 9.89 | 58.26 |
| w2g64 | C4 | 7.87 | 9.30 | 59.24 |
| w2g64 | Redpajama | 7.70 | 9.36 | 58.99 |

## F    ADDITIONAL ABLATION ANALYSIS

**Quantization Group Size.** The group size is a crucial hyperparameter in weight-only quantization. A smaller group size offers more granular compression and reduces quantization loss but increases the number of quantization parameters required. As indicated in Table 12, a group size of 64 strikes an optimal balance for 2-bit quantization using EfficientQAT. It outperforms a group size of 128 by achieving a 0.31 lower perplexity and a 0.64% higher accuracy, yet it slightly underperforms compared to a group size of 32, with a marginal difference of 0.09 in perplexity and 0.14% in accuracy.

**Training Dataset.** More trainable parameters can increase the risk of overfitting. Previous works (Gong et al., 2024) show that a similar distribution between the calibration dataset and the test dataset can improve test accuracy. RedPajama and C4 datasets are diverse, while WikiText2 is simpler and sourced from Wikipedia. The close distribution of training and test datasets for WikiText2 results in significantly lower WikiText2 perplexity when using it as a calibration dataset. However, the average accuracy of zero-shot tasks in Table R7 shows that Block-AP's generation ability is excellent, with only 0.26% and 1.28% accuracy declines when changing the calibration dataset from RedPajama to WikiText2 for w3g128 and w2g64, respectively. Additionally, using C4 as a calibration dataset can even increase the average accuracy by 0.2-0.3 points. Overall, we recommend using Block-AP with more diverse calibration datasets like C4 or RedPajama.

Table 14: **Results about instruction tuning of large vision-language models.** We following the overall training pipeling of LLaVA-1.5 Liu et al. (2023a) and just change the fine-tuning methods. 'QLoRA + Block-AP' indicates that we leverage proposed Block-AP to quantized the QLoRA models into low-bits for fair comparisons. [†] MME's perception scores are normalized to 100 percent.

| Model | Method | #Bit | | MMbench | MME[†] | MM-Vet | ScienceQA | Avg. |
| | | Training | Inference | | | | | |
|---|---|---|---|---|---|---|---|---|
| LLaVA-1.5-7B | LoRA | 16 | 16 | 66.1 | 73.8 | 30.2 | 68.4 | 59.6 |
| | QLoRA | 4+16 | 16 | 64.1 | 72.8 | 30.3 | 68.0 | 58.8 |
| | QLoRA + Block-AP | 4+16 | 4 | 63.6 | 72.0 | 29.8 | 67.7 | 58.3 |
| | EfficientQAT | 4 | 4 | 64.4 | 73.2 | 30.3 | 68.1 | **58.8**(+0.5) |
| | QLoRA + Block-AP | 4+16 | 3 | 62.9 | 71.8 | 29.7 | 66.4 | 57.7 |
| | EfficientQAT | 3 | 3 | 63.2 | 71.4 | 30.9 | 67.3 | **58.2**(+0.5) |
| | QLoRA + Block-AP | 4+16 | 2 | 53.7 | 64.3 | 28.9 | 60.7 | 51.9 |
| | EfficientQAT | 2 | 2 | 62.3 | 68.0 | 27.8 | 63.4 | **55.4**(+3.5) |
| LLaVA-1.5-13B | LoRA | 16 | 16 | 68.5 | 77.1 | 38.3 | 71.2 | 63.8 |
| | QLoRA | 4+16 | 16 | 67.6 | 76.9 | 36.0 | 69.9 | 62.7 |
| | QLoRA + Block-AP | 4+16 | 4 | 67.4 | 76.6 | 35.6 | 69.3 | 62.4 |
| | EfficientQAT | 4 | 4 | 67.5 | 74.8 | 35.6 | 70.2 | 62.0(-0.4) |
| | QLoRA + Block-AP | 4+16 | 3 | 66.8 | 75.5 | 34.5 | 68.4 | 61.3 |
| | EfficientQAT | 3 | 3 | 67.4 | 74.8 | 35.3 | 69.3 | **61.7**(+0.4) |
| | QLoRA + Block-AP | 4+16 | 2 | 62.5 | 72.1 | 32.5 | 65.0 | 58.0 |
| | EfficientQAT | 2 | 2 | 63.9 | 73.1 | 33.9 | 68.6 | **59.9**(+1.9) |

## G    INSTRUCTION TUNING FOR LVLMS.

Traditional Q-PEFT methods only do experiments on the language models. In this section, we further extend proposed EfficientQAT into Large vision-Language models (LVLMs) such as LLaVA (Liu et al., 2023b).

**Training and Evaluation.** For the fine-tuning of large vision-language models (LVLMs), we largely align with LLaVA1.5 (Liu et al., 2023a), which encompass the training model, datasets, and hyperparameters[8]. Unlike LLaVA1.5, which begins fine-tuning with full-precision Vicuna models using either full fine-tuning or LoRA-based methods (Hu et al., 2021), EfficientQAT starts with Vicuna models already quantized using our Block-AP method and continues with our E2E-QP fine-tuning approach. The training process involves two steps: initially freezing the LLM and pre-training

---

[8]For comprehensive details, please consult the official repository at https://github.com/haotian-liu/LLaVA.

a projector to align features with a Vision Transformer (ViT), followed by end-to-end fine-tuning of both the LLM and the projector. For EfficientQAT, we modify the learning rates in the second step to $2 \times 10^{-5}$ for 4-bit and $3 \times 10^{-5}$ for 2-bit and 3-bit.

**Evaluation.** Evaluation of the fine-tuned LVLMs are conducted across four benchmarks: MME (Fu et al., 2023), MM-Vet (Yu et al., 2023), MMBench (Liu et al., 2023d), and ScienceQA (Lu et al., 2022).

**Baseline.** We compare our results with those of QLoRA (Dettmers et al., 2023a), applying our Block-AP method to quantize the QLoRA fine-tuned models to low bits for fair comparison.

**Results.** As shown in Table 14, EfficientQAT outperforms QLoRA (Dettmers et al., 2023a) in low-bit settings for both LLaVA-1.5-7B and LLaVA-1.5-13B models, consistent with previous results in LMMs. Remarkably, the 2-bit LLaVA-1.5-13B model trained with EfficientQAT achieves an average score of 59.9, surpassing the 59.6 of the FP16 LLaVA-1.5-7B model trained with LoRA. However, there is a slight performance decrease observed in the 4-bit EfficientQAT and 16-bit QLoRA compared to the 16-bit LoRA, indicating that further research is needed to optimize Q-PEFT within LVLMs.

## H    FULL RESULTS

In Table 1, we present the average accuracy for five zero-shot tasks. This section offers a detailed breakdown of the task-specific accuracy numbers. Specifically, Tables 15, 16, and 17 detail the performance of 4-bit, 3-bit, and 2-bit quantization, respectively.

Table 15: 4-bit Llama 2 & 3 zero-shot accuracy by lm_eval v0.4.2 ( acc is reported, not acc_norm )

| Model | Method | Bits | Group | WinoGrande | HellaSwag | ArcC | ArcE | PiQA | Average accuracy↑ |
|---|---|---|---|---|---|---|---|---|---|
| | - | - | 16 | 69.22 | 57.16 | 43.52 | 76.26 | 78.07 | 64.85 |
| | RTN | 4 | 128 | 68.35 | 56.91 | 43.52 | 76.26 | 77.58 | 64.52 |
| | GPTQ | 4 | 128 | 69.06 | 56.36 | 42.15 | 75.63 | 78.02 | 64.24 |
| 2-7B | AWQ | 4 | 128 | 68.98 | 56.40 | 43.86 | 76.14 | 77.31 | 64.54 |
| | OmniQ | 4 | 128 | 68.98 | 56.59 | 43.34 | 75.76 | 77.91 | 64.52 |
| | AutoRound | 4 | 128 | 68.67 | 56.79 | 42.58 | 75.76 | 78.13 | 64.39 |
| | QuIP# | 4 | - | 69.22 | 56.65 | 43.00 | 75.51 | 78.02 | 64.48 |
| | EfficientQAT | 4 | 128 | 69.22 | 57.00 | 42.32 | 75.13 | 77.69 | 64.27 |
| | - | 16 | - | 72.22 | 60.07 | 48.29 | 79.42 | 79.05 | 67.81 |
| | RTN | 4 | 128 | 72.14 | 59.77 | 47.95 | 79.00 | 78.62 | 67.50 |
| | GPTQ | 4 | 128 | 72.14 | 59.76 | 47.53 | 78.58 | 78.35 | 67.27 |
| 2-13B | AWQ | 4 | 128 | 73.32 | 59.80 | 46.50 | 79.38 | 79.05 | 67.61 |
| | OmniQ | 4 | 128 | 72.06 | 59.53 | 47.18 | 78.37 | 78.35 | 67.10 |
| | AutoRound | 4 | 128 | 71.67 | 59.87 | 47.01 | 79.25 | 79.00 | 67.36 |
| | QuIP# | 4 | - | 72.69 | 59.49 | 46.59 | 78.62 | 79.00 | 67.28 |
| | EfficientQAT | 4 | 128 | 71.98 | 59.87 | 47.53 | 79.34 | 78.89 | 67.52 |
| | - | 16 | - | 77.98 | 64.77 | 54.44 | 82.70 | 82.15 | 72.41 |
| | RTN | 4 | 128 | 78.14 | 63.93 | 54.78 | 82.79 | 81.66 | 72.26 |
| | GPTQ | 4 | 128 | 78.22 | 64.45 | 54.61 | 82.79 | 81.88 | 72.39 |
| 2-70B | AWQ | 4 | 128 | 77.58 | 64.48 | 55.12 | 82.70 | 82.32 | 72.44 |
| | OmniQ | 4 | 128 | 77.51 | 64.52 | 55.12 | 82.91 | 81.88 | 72.39 |
| | AutoRound | 4 | 128 | 78.30 | 64.60 | 54.52 | 82.87 | 82.05 | 72.47 |
| | QuIP# | 4 | - | 77.82 | 64.51 | 54.44 | 82.37 | 81.72 | 72.17 |
| | EfficientQAT | 4 | 128 | 78.45 | 64.57 | 55.12 | 83.00 | 81.94 | 72.62 |
| | - | - | 16 | 72.61 | 60.17 | 50.43 | 80.09 | 79.60 | 68.58 |
| | RTN | 4 | 128 | 73.16 | 59.04 | 48.38 | 79.25 | 79.11 | 67.79 |
| 3-8B | GPTQ | 4 | 128 | 73.72 | 59.17 | 47.78 | 79.38 | 78.94 | 67.80 |
| | AWQ | 4 | 128 | 73.01 | 59.43 | 50.00 | 79.55 | 79.22 | 68.24 |
| | EfficientQAT | 4 | 128 | 72.53 | 59.43 | 50.94 | 79.84 | 79.43 | 68.43 |
| | - | 16 | | 80.51 | 66.36 | 60.41 | 86.99 | 82.37 | 75.33 |
| | RTN | 4 | 128 | 78.77 | 65.83 | 57.34 | 85.69 | 82.26 | 73.98 |
| 3-70B | GPTQ | 4 | 128 | 80.51 | 66.12 | 59.04 | 85.77 | 82.26 | 74.74 |
| | AWQ | 4 | 128 | 80.35 | 65.82 | 59.13 | 86.41 | 82.15 | 74.77 |
| | EfficientQAT | 4 | 128 | 79.24 | 66.27 | 59.13 | 85.86 | 82.37 | 74.57 |

Table 16: 3-bit Llama 2 & 3 zero-shot accuracy by lm_eval v0.4.2 ( acc is reported, not acc_norm )

| Model | Method | Bits | Group | WinoGrande | HellaSwag | ArcC | ArcE | PiQA | Average accuracy↑ |
|-------|--------|------|-------|-----------|-----------|------|------|------|-------------------|
| 2-7B | - | - | 16 | 69.22 | 57.16 | 43.52 | 76.26 | 78.07 | 64.85 |
| | RTN | 3 | 128 | 67.56 | 54.90 | 38.57 | 72.98 | 76.28 | 62.06 |
| | GPTQ | 3 | 128 | 68.59 | 53.66 | 40.19 | 73.74 | 76.01 | 62.44 |
| | AWQ | 3 | 128 | 67.40 | 54.98 | 41.64 | 74.07 | 76.01 | 62.82 |
| | OmniQ | 3 | 128 | 66.69 | 54.42 | 39.85 | 74.37 | 76.77 | 62.42 |
| | AutoRound | 3 | 128 | 68.27 | 55.33 | 42.92 | 75.25 | 76.82 | 63.72 |
| | QuIP# | 3 | - | 68.19 | 55.85 | 41.89 | 74.62 | 77.04 | 63.52 |
| | EfficientQAT | 3 | 128 | 69.14 | 55.90 | 42.83 | 74.66 | 77.58 | 64.02 |
| 2-13B | - | 16 | - | 72.22 | 60.07 | 48.29 | 79.42 | 79.05 | 67.81 |
| | RTN | 3 | 128 | 70.72 | 57.74 | 44.62 | 77.69 | 78.07 | 65.77 |
| | GPTQ | 3 | 128 | 70.88 | 57.83 | 45.65 | 77.99 | 78.56 | 66.18 |
| | AWQ | 3 | 128 | 71.82 | 58.58 | 44.62 | 77.95 | 77.75 | 66.14 |
| | OmniQ | 3 | 128 | 70.01 | 58.46 | 46.16 | 77.86 | 78.40 | 66.18 |
| | AutoRound | 3 | 128 | 71.59 | 59.11 | 45.82 | 78.58 | 78.29 | 66.68 |
| | QuIP# | - | 3 | 72.45 | 58.26 | 44.62 | 77.90 | 78.07 | 66.26 |
| | EfficientQAT | 3 | 128 | 72.06 | 59.01 | 47.95 | 79.00 | 78.40 | 67.28 |
| 2-70B | - | 16 | - | 77.98 | 64.77 | 54.44 | 82.70 | 82.15 | 72.41 |
| | RTN | 3 | 128 | 77.90 | 61.98 | 52.39 | 81.10 | 80.79 | 70.83 |
| | GPTQ | 3 | 128 | 77.66 | 62.94 | 53.67 | 81.65 | 81.45 | 71.47 |
| | AWQ | 3 | 128 | 76.48 | 63.75 | 53.67 | 81.40 | 81.77 | 71.41 |
| | OmniQ | 3 | 128 | 76.48 | 63.54 | 52.82 | 81.02 | 81.50 | 71.07 |
| | AutoRound | 3 | 128 | 76.56 | 63.83 | 52.56 | 81.73 | 81.50 | 71.24 |
| | QuIP# | 3 | - | 76.24 | 64.22 | 55.89 | 82.11 | 82.21 | 72.13 |
| | EfficientQAT | 3 | 128 | 77.27 | 64.20 | 53.75 | 81.73 | 81.83 | 71.76 |
| 3-8B | - | - | 16 | 72.61 | 60.17 | 50.43 | 80.09 | 79.60 | 68.58 |
| | RTN | 3 | 128 | 66.54 | 50.87 | 36.69 | 65.36 | 74.16 | 58.72 |
| | GPTQ | 3 | 128 | 70.88 | 55.13 | 37.80 | 65.24 | 73.83 | 60.58 |
| | AWQ | 3 | 128 | 70.96 | 55.43 | 44.20 | 75.84 | 77.69 | 64.82 |
| | EfficientQAT | 3 | 128 | 71.51 | 57.81 | 48.81 | 80.01 | 78.63 | 67.35 |
| 3-70B | - | 16 | | 80.51 | 66.36 | 60.41 | 86.99 | 82.37 | 75.33 |
| | RTN | 3 | 128 | 65.90 | 54.22 | 48.46 | 78.83 | 79.05 | 65.29 |
| | GPTQ | 3 | 128 | 78.14 | 62.58 | 52.99 | 82.07 | 80.63 | 71.28 |
| | AWQ | 3 | 128 | 78.85 | 64.26 | 58.36 | 84.51 | 82.26 | 73.65 |
| | EfficientQAT | 3 | 128 | 77.82 | 65.53 | 55.12 | 83.12 | 80.52 | 72.42 |

Table 17: 2-bit Llama 2 & 3 zero-shot accuracy by lm_eval v0.4.2 ( acc is reported, not acc_norm )

| Model | Method | Bits | Group | WinoGrande | HellaSwag | ArcC | ArcE | PiQA | Average accuracy↑ |
|---|---|---|---|---|---|---|---|---|---|
| | - | - | 16 | 69.22 | 57.16 | 43.52 | 76.26 | 78.07 | 64.85 |
| | GPTQ | 2 | 128 | 55.17 | 32.59 | 21.25 | 40.45 | 58.32 | 41.56 |
| | OmniQ | 2 | 128 | 55.88 | 40.28 | 23.46 | 50.13 | 65.13 | 46.98 |
| | AutoRound | 2 | 128 | 61.01 | 40.28 | 32.25 | 65.99 | 72.96 | 54.50 |
| 2-7B | AQLM | 2 | 2x8 | 65.27 | 49.96 | 32.85 | 66.92 | 73.07 | 57.61 |
| | AQLM | 2 | 1x16 | 65.19 | 53.42 | 39.68 | 74.07 | 76.88 | 61.85 |
| | QuIP# | 2 | - | 65.67 | 52.19 | 37.88 | 71.84 | 75.46 | 60.61 |
| | EfficientQAT | 2 | 128 | 66.22 | 50.84 | 36.52 | 69.78 | 74.16 | 59.50 |
| | EfficientQAT | 2 | 64 | 65.98 | 51.58 | 36.86 | 70.96 | 75.30 | 60.14 |
| | - | 16 | - | 72.22 | 60.07 | 48.29 | 79.42 | 79.05 | 67.81 |
| | GPTQ | 2 | 128 | 55.80 | 41.06 | 21.93 | 55.60 | 67.08 | 48.29 |
| | OmniQ | 2 | 128 | 57.93 | 46.23 | 30.29 | 63.22 | 70.13 | 53.56 |
| | AutoRound | 2 | 128 | 64.33 | 53.35 | 38.57 | 71.17 | 76.17 | 60.72 |
| 2-13B | AQLM | 2 | 2x8 | 66.22 | 54.62 | 40.10 | 73.06 | 77.09 | 62.22 |
| | AQLM | 2 | 1x16 | 70.09 | 57.62 | 43.52 | 75.25 | 78.29 | 64.95 |
| | QuIP# | 2 | - | 69.06 | 56.53 | 42.92 | 75.72 | 77.97 | 64.44 |
| | EfficientQAT | 2 | 128 | 68.90 | 55.66 | 42.83 | 75.04 | 76.99 | 63.88 |
| | EfficientQAT | 2 | 64 | 68.36 | 55.27 | 41.89 | 74.83 | 77.04 | 63.48 |
| | - | 16 | - | 77.98 | 64.77 | 54.44 | 82.70 | 82.15 | 72.41 |
| | GPTQ | 2 | 128 | 49.57 | 25.04 | 22.70 | 25.08 | 49.51 | 34.38 |
| | OmniQ | 2 | 128 | 64.33 | 35.45 | 33.28 | 67.21 | 74.10 | 54.87 |
| | AutoRound | 2 | 128 | 74.90 | 59.65 | 46.59 | 78.37 | 79.00 | 67.70 |
| 2-70B | AQLM | 2 | 2x8 | 75.61 | 61.94 | 51.45 | 79.76 | 80.47 | 69.85 |
| | AQLM | 2 | 1x16 | 76.01 | 62.78 | 52.99 | 81.36 | 81.07 | 70.84 |
| | QuIP# | 2 | - | 75.77 | 62.86 | 52.65 | 81.90 | 81.39 | 70.91 |
| | EfficientQAT | 2 | 128 | 73.64 | 61.58 | 49.23 | 80.01 | 80.20 | 68.93 |
| | EfficientQAT | 2 | 64 | 74.59 | 61.78 | 50.77 | 80.13 | 80.14 | 69.48 |
| | - | - | 16 | 72.61 | 60.17 | 50.43 | 80.09 | 79.60 | 68.58 |
| 3-8B | AQLM | 2 | 1x16 | 71.82 | 55.44 | 41.21 | 74.24 | 77.80 | 64.10 |
| | EfficientQAT | 2 | 128 | 65.67 | 50.74 | 36.01 | 69.15 | 75.30 | 59.37 |
| | EfficientQAT | 2 | 64 | 67.72 | 51.86 | 37.03 | 71.17 | 76.03 | 60.76 |
| | - | 16 | | 80.51 | 66.36 | 60.41 | 86.99 | 82.37 | 75.33 |
| 3-70B | AQLM | 2 | 1x16 | 78.22 | 63.47 | 50.34 | 78.83 | 79.65 | 70.10 |
| | EfficientQAT | 2 | 128 | 69.46 | 60.75 | 48.81 | 79.25 | 79.60 | 67.57 |
| | EfficientQAT | 2 | 64 | 74.03 | 61.60 | 49.06 | 77.40 | 77.37 | 67.89 |

