# OpenReview forum: "EfficientQAT: Efficient Quantization-Aware Training for Large Language Models"
_ICLR.cc/2025/Conference — Submitted to ICLR 2025_

### Official Review · Reviewer_u4Hv · 2024-10-29

**Soundness:** 2
**Presentation:** 2
**Contribution:** 1
**Rating:** 3
**Confidence:** 4

**Summary:**

Although quantization-aware training (QAT) offers a solution by reducing memory consumption through low-bit representations with minimal accuracy loss, it is impractical due to substantial training resources. To address this, this paper proposes Efficient Quantization-Aware Training (EfficientQAT), a more feasible QAT algorithm. EfficientQAT involves two consecutive phases: Block-wise training of all parameters (Block-AP) and end-to-end training of quantization parameters (E2E-QP). Block-AP enables direct training of all parameters in a block-wise manner, reducing accuracy loss in low-bit scenarios. E2E-QP then trains only the quantization parameters (step sizes) end-to-end, further improving the performance of quantized models by considering interactions among all sub-modules.

**Strengths:**

Extensive experiments demonstrate that EfficientQAT outperforms previous quantization methods across a range of models, including
base LLMs, instruction-tuned LLMs, and multimodal LLMs, with scales from 7B to 70B parameters at various quantization bits. For instance, EfficientQAT obtains a 2-bit Llama-2-70B model on a single A100-80GB GPU in 41 hours, with less than 3 points accuracy degradation compared to the full precision (69.48 vs. 72.41).

**Weaknesses:**

The novelty of the proposed method may be limited. To reduce the training cost, this paper generally only trains part of the model during quantization and freeze other parts, thus saving memory. This idea is straightforward and has been investigated in other works, such as (Li et al., 2021; Shao et al., 2023). The quantization method generally follows very traditional QAT method to learn model weights and quantization parameters. The technical contribution may be limited.

The comparisons of training time with existing methods in table 9 does not seem to be solid. Training time is determined by multiple factors such as algorithms and training data. Typically, it takes more time to train if using more training data. To make a fair comparison, it is better to try using the same amount of data for training of all methods.   If it uses less training time because of less training data, it is hard to say that it is more training efficient. It may be better to discuss this training data issue in training time comparison.

As this paper proposes to train all parameters in a block-wise manner, a more direct baseline is to train the full model with all blocks during quantization. It is better to compare with this baseline to demonstrate the training efficiency such as final accuracy and training time.  For example, as the proposed method only trains one block in a time, and it needs to train multiple rounds as the model has multiple blocks, does it really use less training time compared with training all blocks in one round? And what is the PPL or accuracy performance compared with training all blocks in one round? The current baselines does not seem to cover this baseline. The LLM-QAT adopts knowledge distillation, which is different from the setting in this paper. It is better to discuss the comparison with the straightforward baseline to train all blocks.

The baselines use various finetuning or calibration datasets or training settings, such as C4 for GPTQ, Pile for AWQ, and so on. It is hard to say whether the performance difference is introduced by the proposed method or the different finetuning dataset or settings. It is better to provide more discussion for this dataset or setting issue during quantization.

AWQ in table 1 can perform better than the proposed method in some cases. It is better to discuss this issue. As a post training quantization method, AWQ typically costs less resource than QAT methods. It is better to discuss why it can lead to a better performance.

**Questions:**

See the weakness.

It may be better to discuss this training data issue in training time comparison.

It is better to discuss the comparison with the straightforward baseline to train all blocks.

It is better to provide more discussion for this dataset or setting issue during quantization.

---

### Official Review · Reviewer_14pf · 2024-10-31

**Soundness:** 2
**Presentation:** 2
**Contribution:** 2
**Rating:** 3
**Confidence:** 5

**Summary:**

This paper proposes an efficient Quantization-Aware Training (QAT) method for LLMs.
In detail, the paper introduces the block-wise weight-only QAT (Block-AP) to reduce the memory cost during training and further optimization with the training of scale in weight quantizers (E2E-QP).
The experiments show that the proposed QAT method can achieve better performance than previous quantization works.
For example, a 2-bit Llama-2-70B model trained on a single A100-80GB GPU with EfficientQAT.

**Strengths:**

1. Block-wise QAT reduces the GPU memory requirement and total training time.
2. The method performs well on small models (LLaMA-2-7B and LLaMA-3-8B) with lower than 4 bits.

**Weaknesses:**

1. The novelty of this paper is limited. The block-wise QAT is not novel as the block-wise methods are commonly used in quantization and pruning methods for LLMs. For example, [1] [2] [3] [4] [5] [6] adopted layerly strategy in their works, the famous one is GPTQ [2].
2. The work focuses on the weight-only quantization, while the comparison works contain many weight and activation both quantized methods including OmniQuant and LLM-QAT, which shows unfairness.
3. The work did not achieve good results with 2,3,4-bit weight quantization with LLaMA2 and LLaMA3 according to Table 1. For example, for uniform quantization, the paper did not achieve better results with 4-bit on LLaMA-2-7B, LLaMA-2-13B, and LLaMA-3-70B. And with 3-bit on LLaMA-3-70B, the paper even performs worse than AWQ which is a post-training quantization method (where even denotes their results in bold).
4. The paper only includes the comparison with QLoRA (with GPTQ weights), QA-LoRA, IR-QLoRA and PEQA methods in Table 4, while not include these methods in the main results table.
5. This paper adopts 4096 samples in RedPajama datasets with 2048 sequence length for the Block-AP and 4096 sequence length for the E2E-QP. Thus, the comparison with those PTQ works (AWQ, OminiQuant and GPTQ) is unfair. The paper did not explain the setting of those PTQ works, if they are also use such amount of data with such sequence length for calibration? As according to the Figure 3, the proposed method is sensitive to the number of samples range from 128 to 4096, while the GPTQ only adopts 128 calibration samples from the training dataset of Wiki or C4 in their original setting. Besides, according to Table 13, the proposed method performs worse when adopting Wiki or C4 as training dataset.
6. As for the training time, the paper should include the LoRA-based methods for comparison including those in Table 4: QLoRA (with GPTQ weights), QA-LoRA, IR-QLoRA and PEQA. Also, the post-training quantization methods are also needed to be included.
7. The quantization overhead for LLMs mainly caused from the activation quantization, which this paper did not take into consideration, even the 16 bit activation results are not included.
8. The ablation for scale optimization (E2E-QP) with weights from post-training quantization methods compared to Block-AP weights is needed.


[1] Layer-Wise Quantization: A Pragmatic and Effective Method for Quantizing LLMs Beyond Integer Bit-Levels \
[2] GPTQ: Accurate Post-Training Quantization for Generative Pre-trained Transformers \
[3] Streamlining Redundant Layers to Compress Large Language Models \
[4] Compressing Large Language Models by Streamlining the Unimportant Layer \
[5] Layer-Wise Quantization: A Pragmatic and Effective Method for Quantizing LLMs Beyond Integer Bit-Levels \
[6] Sheared LLaMA: Accelerating Language Model Pre-training via Structured Pruning

**Questions:**

1. How many samples and what kind of datasets are used for the calibration of those post-training quantization methods? What is the detailed experiment setup for other methods?
2. The zero-shot accuracy results and ppl results for the LoRA based methods including: QLoRA (with GPTQ weights), QA-LoRA, IR-QLoRA and PEQA.
3. How about the results with 16 bit activation quantization?
4. Although this work adopts block-wise QAT method, this work still adopts the full parameter fine-tuning, which costs more resource compared to LoRA. Meanwhile, the LoRA based methods can optimize the model globally, while the proposed method can only optimize the model within blocks (although the further optimization of scales is global). Thus, the question is that, the optimization is brought by the Block-AP or the E2E-QP? What if directly using GPTQ, AWQ or QA-LoRA (or other LoRA based methods) weights and using E2E-QP for further optimization?

---

### Official Review · Reviewer_6j3y · 2024-11-04

**Soundness:** 2
**Presentation:** 2
**Contribution:** 2
**Rating:** 3
**Confidence:** 4

**Summary:**

The paper proposes EfficientQAT, a novel quantization-aware training (QAT) framework tailored for large language models (LLMs). Aiming to address the high memory and computational demands of traditional QAT, EfficientQAT introduces a two-phase approach: Block-wise training of all parameters (Block-AP) and end-to-end training of quantization parameters (E2E-QP). Block-AP enables training of all parameters within each block, increasing flexibility and optimization efficiency, while E2E-QP further enhances model performance by training quantization parameters across all blocks. Experimental results demonstrate that EfficientQAT outperforms existing quantization methods in accuracy and memory efficiency across LLMs of varying sizes, from 7B to 70B parameters, at low-bit settings.

**Strengths:**

1. This work adopts a two-phase approach to effectively minimizes accuracy loss even at lower bit levels, which is novel. It is the first method to directly train all parameters in a block-wise fashion, minimizing accuracy loss in low-bit settings by expanding the solution space for optimization. Following this, E2E-QP focuses solely on training the quantization parameters (step sizes) in an end-to-end manner, enhancing the performance of quantized models by accounting for interactions across all sub-modules.

2. Great performance. In terms of training speed, EfficientQAT can obtain a 2-bit Llama-2-70B model on a single A100-80GB GPU in 41 hours with less low accuracy degradation, getting better acceleration performance than other baseline.

**Weaknesses:**

1. Not enough novelty. The main contribution appears to be the proposed training pipeline, but this pipeline does not introduce substantial advancements beyond existing techniques. While the combination of block-wise and end-to-end training is interesting, it builds on straightforward adaptations of known methods rather than providing an innovative or fundamentally new approach.

2. Unfair comparison. The proposed method only did weight quantization, but many of baselines were using both activation and weight quantization.

3. Performance is not good enough compared to baselines. For example, in Table 3, the performance of the proposed method is worse than QuIP\# and AQLM almost in all settings. Also in Table 15, in model 2-7B, the accuracy of the EfficientQAT is the lowest among all methods.

**Questions:**

In the baselines, are they using the same sampling numbers as EfficientQAT?

---

### Meta-Review · Area_Chair_u1eB · 2024-12-14

**Metareview:**

This paper introduces a quantization-aware training framework called EfficientQAT designed for large language models. The method is two-phase and combines block-wise training and end-to-end tuning of quantization parameters. While the idea is practical and shows promise in improving training efficiency and memory usage, it doesn’t bring much novelty, as block-wise methods are already well-explored in the field. The reviewers also raise concerns about comparisons with other methods, for example, the baselines often involve both weight and activation quantization, while this work focuses only on weight quantization, making the evaluations feel uneven. Moreover, the method underperforms in several cases against existing techniques like AWQ and OmniQuant, and key ablation studies, such as applying E2E-QP to other baselines, are missing. The authors didn't provide rebuttal, so these concerns remain unaddressed. Overall, while the approach has some merit, the limited innovation and uneven comparisons make it hard to recommend for acceptance.

**Additional Comments On Reviewer Discussion:**

The reviewers raised concerns about limited novelty, unfair comparison and key ablation studies missing. However, the authors didn't provide rebuttal, so these concerns remain unaddressed.

---

### Decision · Program_Chairs · 2025-01-22

Reject